# Blockchain for video watermarking: An enhanced copyright protection approach for video forensics based on perceptual hash function

**Saad Mohamed Darwish[ID][1]\*, Mona Mahamod Abu-Deif[2], Saleh Mesbah Elkaffas[3]**

**1** Department of Information Technology, Institute of Graduate Studies and Research, Alexandria University, Alexandria, Egypt, **2** Full Stack Web Developer (e-space), Alexandria, Egypt, **3** College of Computing and Information Technology, Arab Academy for Science, Technology and Maritime Transport, Alexandria, Egypt

\* saad.darwish@alex.edu.eg

**Data Availability Statement:** Data is available on Middlebury Stereo Vision Datasets, a public benchmark dataset available at https://vision.middlebury.edu/stereo/data. The authors confirm

## Abstract

As a direct result of advancements in digital technology and the Internet, the copyright protection and information integrity of multimedia that are being published across the Internet have emerged as a major and urgent issue that needs to be addressed. The technique of digital watermarking may be used to protect intellectual property. In terms of authentication, resilience, storage, and capacity of digital watermarking information, there is still room for development. Blockchain's potential in video copyright protection and management applications has motivated researchers. Copyright owners and consumers may now communicate directly via copyright protection apps built on the blockchain, eliminating the need for distributers and the associated fees. Nonetheless, the current blockchain–based video watermarking solutions require storing a significant number of coordinates depending on the watermark size and are susceptible to video frame attacks on the video frame texture region. This study proposes an enhanced video copyright management approach that incorporates digital watermarking, the blockchain, and a perceptual hash function. Watermark information is stored on a blockchain structure, which also acts as a timestamp for verification purposes. To verify watermark data without the source video, a perceptual hash function is employed to compute a hash value based on the structural information of video frames. The contribution is in learning how to extract a hash function from a small number of video frames that still adequately represent a large amount of video content while also reducing the number of unnecessary video frames and the amount of computation required to summarize and index that content in a blockchain. This expedites the dissemination of copyrighted works and increases their security and readability, hence facilitating their circulation on the Internet. Our experimental results demonstrate that this approach is memory efficient, as it only needs to store one key for each key frame, regardless of the size of the watermark. Additionally, the overall robustness is greatly improved by using the blockchain's random hash function. Therefore, new and important advancements in video watermarking have been realized because of this effort.

others would be able to access these data in the same manner as the authors and that the authors did not have any special access privileges that others would not have.

**Funding:** The author(s) received no specific funding for this work.

**Competing interests:** The authors have declared that no competing interests exist.

# I. Introduction

Copyright violations in the video medium are common because of the nature of the medium itself, which makes it both simple to alter and difficult to differentiate. Massive amounts of video data are sent across the network and may be readily manipulated by tactics including copying, tampering, and redistributing. You can get pirated video materials just about everywhere online. This has a devastating effect on the economy as it slows down the commercialization of the video industry. Video works have both opportunities and challenges brought about by the Internet. On one hand, there is the issue of industrial copyright disorder, which poses a difficulty in this field. The failure to address this issue in a timely manner may impede progress in the copyright market for online video productions. It is imperative that immediate action be taken to ensure the integrity of audiovisual content via copyright protection [1–3]. Fig 1 illustrates how a video watermark may be used for copyright protection [4].

In the last 20 years, technology has advanced enough to the point that video watermarking is now a viable option. For video watermarking to work, it must be both undetectable and resilient. The former requires both objective and subjective quality between the watermarked and original movies. Regardless of whether the watermarked video is altered or destroyed by an outside party, the owner may simply re-establish the embedded watermark. [5,6]. There are two distinct categories of video watermarking technologies: active and passive. Digital watermarks, fingerprints, and digital signatures are examples of active techniques that rely on preexisting information, such as hash values or signatures, as evidence [7]. The passive technique, on the other hand, relies on compression artefacts and noise residue in the modified video and is thus a "blind" form of tamper detection that does not rely on knowledge of the original contents. Because most security cameras and surveillance systems already have video compression techniques built in, forging is often accomplished by decompressing the original compressed video, making changes, then recompressing it [7].

The following are issues with conventional video digital watermarking technology: To begin, the current methods have a low level of resilience. There are many different types of video attacks, including frame average attacks, frame deletion attacks, and frame reassembly attacks. The video watermarking system, therefore, has to be secure against these threats. Additionally, there is still room for development in making geometric attacks, compression attacks, and other types of attacks more resistant to common countermeasures. Secondly, the robustness and the vulnerability of a video single watermarking technique are mutually limiting. In the event of tampering, the fragile watermarking can pinpoint and measure the exact location

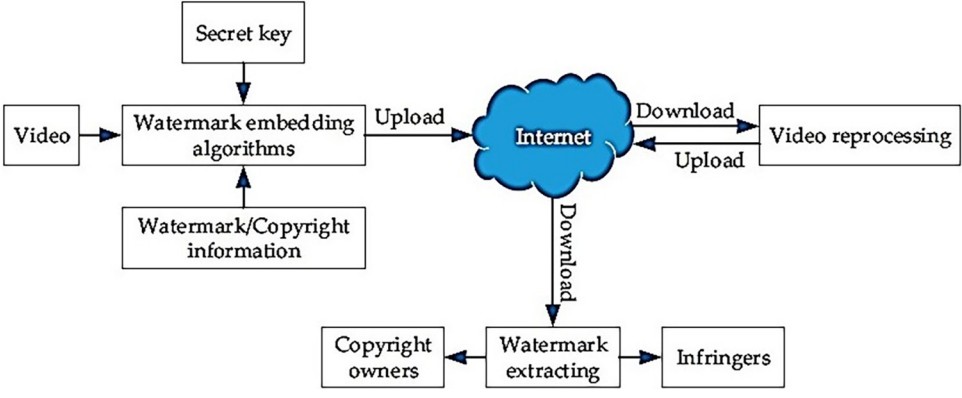

**Fig 1. The application of video watermark in copyright protection.**

of the alteration. What's more, it's fairly uncommon for a single watermarking system to be too weak to prevent unauthorized access [5,6].

Thirdly, conventional digital copyright protection methods rely on centralized databases. This means that data security is quite vulnerable. Today, due to the Internet's explosive growth, copyright protection is more important than ever, yet centralized databases can't keep up with the demand. Fourthly, the watermark does a poor job of identifying its owner. It is common practice for people or organizations to embed the watermark of important information with assurance and privacy into the protected resources they want to protect. A watermark's usefulness to its owner is diminished if it is not notarized by a credible third-party certifying body. This makes it impossible to verify the watermark's claimed connection to a specific person or organization. Verification by an outside certifying body is often a time-consuming and expensive process [1,7].

Blockchain's key features—transparency, decentralization, trustworthy databases, maintenance, trackability, security, and programmable contracts—offer novel approaches to the problems of digital intellectual property protection and traceability. Copyright protection in the digital realm works well with blockchain technology. To begin with, data identification and explicit ownership of rights are made possible by the blockchain because of the permanent connection that can be established between users and data objects. Second, the preservation of the blockchain and the impossibility of tampering or forging any of the data stored within may give irrefutable proof of the authenticity of digital works and strengthen the case for their integrity [1–4]. However, a decentralized system and a proof-of-work algorithm are necessary for a realistic blockchain implementation. Integrating a full-principle implementation might be challenging in certain contexts due to factors including a lack of infrastructure, extremely sensitive data, or low power devices [8–10]. Existing blockchain-based video watermarking methods are vulnerable to video frame attacks and require storing a large number of coordinates depending on the watermark size [2,6,7].

In this research, we come up with a new strategy for integrating the blockchain's aforementioned properties into a centralized database for video copyright protection systems. The video feature's validity is ensured by the content-based video hashing where a normalized vector representation of the video is utilized and stored to each block (watermark information), which eliminates the risk of an attacker recalculating the hash value and thereby altering the whole database. To facilitate fast content summarization of video and reduce the size of the stored watermark, the proposed strategy constructs a perceptual hash function on the basis of a video key frame extraction technique that employs as few video frames as feasible to represent as much video information as possible, reduces redundant video frames, and reduces the amount of computation.

The suggested method actively generates an integrity value for every video segment and checks its validity using these values, making it able to resist attacks on the video frame texture area. Furthermore, it provides more robust integrity verification against intrinsic manipulations or tampering since it does not require the discovery of inconsistent periodic artefacts and noisy residues in the video for tamper identification. The contributions of the proposed method are summarized as follows:

- The concept of a blockchain is presented in the context of centralized video storage. To the best of our knowledge, this is the first study to address how a blockchain hash value (perceptual hash function) for centralized video data is generated. To get this hash value, we use an efficient technique for extracting video key frames that uses the fewest possible frames while yet adequately expressing all of the necessary video data.

- Our blockchain-wise approach can achieve faster processing and less memory consumption for video copyright protection as the computation of hash values (stored in the blockchain as watermark information) helps cut down on unnecessary video frames and computational overhead.

- A variety of security issues and attacks are rigorously investigated, so assuring that the recommended technique is highly secure and applicable to real-world applications.

The rest of the paper is organized as follows. In Section II, we briefly review the state-of-the-art video watermarking techniques. The proposed method is presented in Section III. The experimental results and analysis are demonstrated in Section IV and finally, the paper is concluded in Section V.

## II. Related work

Using blockchain technology for preservation of intellectual property rights has been a topic of discussion in both business and academics as of late. According to the existing literature, blockchain is a trustworthy and transparent ledger used to address the issues with copyright protection that content owners and producers face. The following section offers a high-level summary of current blockchain-based video content protection systems, including their key features and implementation specifics [11,12].

In [13], the authors suggest a blockchain-based system that ensures the legality of multimedia assets via the use of smart contracts and copyright compliance. To keep track of the specifics of each and every data transaction that is added to the blockchain, the proposed system makes use of a centralized storage solution that is not part of the blockchain itself: the Data Lake. To protect the confidentiality and veracity of the data kept on the data lake, it is encrypted and digitally signed. With the permission of the majority of nodes, only authorized users may access the stored data by verifying their digital signatures and access privileges. This proof-of-concept system for decentralized data management guarantees user privacy and control, but it has not been tested in the real world.

To help satisfy the growing need for an effective method of managing digital rights management on the part of content providers, Kishigami et al. [14] developed a blockchain-based system for managing copyrights on high-definition videos. The suggested strategy is based on the PoW mechanism, which gives the right holders full control over the network. This method encrypts and decrypts the video file's headers at ultra-high resolutions to even out the cryptographic expenses of these operations. However, there is no mechanism in place to reward miners for contributing processing time to the system. It also doesn't let you manage the media file's access policy or display it on different platforms.

A blockchain-based data hiding solution for digital video security is suggested in Reference [15], which enhances integrity authentication of private data and videos. The three components of the suggested procedure are as follows: (1) a data protection management agreement based on a smart contract that consists of registration, inquiry, and transfer contract models; (2) data hiding algorithms that strike a good balance between transparency, embedding capacity, and resilience provide off-chain data protection.; and (3) a procedure to secure data on the blockchain that mostly involves checking the video's integrity and security by adding the content's signature to the blockchain. For multimedia playing to be possible under the proposed method, however, users would have to request data extraction from data-hiding servers.

This study in [16] suggested the use of logo to identify the copyright of sports videos, which can automatically and efficiently determine whether the video is genuine or not, thereby solving the copyright issue of sports videos. This video's authenticity may be confirmed by using

encryption technology to add a logo and logo recognition technology to identify if the logo is missing. In addition, this research offers a logo detection and identification technology based on a convolutional neural network, with the intention of addressing the limitations of current logo recognition technology. By enhancing the classic histogram approach and proposing an analysis algorithm based on a non-uniform block HSV histogram, as well as introducing area weighting factors to identify crucial frames from a video, this technique represents a significant step forward. Then, we utilize a convolutional neural network to pull information from the videos most crucial frames. Once all of the characteristics of the test logo have been compared, the logo detection and identification process will be complete. The results of simulations demonstrate the superiority of convolutional neural network-based feature extraction. The biggest problem with this approach is that it needs a lot of training data and can't handle any video format.

Researchers in [17] looked at the feasibility of using the Bitcoin network to accurately time-stamp video released by a smartphone camera installed in a vehicle in the case of a collision. The data from the camera is sent into the smartphone, where it is hashed and then transferred as a Bitcoin transaction. A hash will be generated and stored on the Bitcoin network, where it will be available to anyone to see but unchangeable by anybody. Proof that the video was not tampered with may be provided by recording a hash of the file in the blockchain of a crypto-currency. Any changes made to the file after the fact would result in a different hash, which would not match the one stored on the blockchain. The downside of this feature is that it is hard to correct a mistake or make any necessary adjustments.

Video files are digitally watermarked by the authors of [18] to identify data manipulation. They start by separating the video, audio, timecodes, and header from a video file. The audio and video are timestamped and watermarked independently using the header and timecodes. The components are watermarked, then assembled for transmission across a specified channel. Watermarking is able to show if data was tampered with in the video or audio format by isolating the two (such as shifted, replaced, or deleted, or if the header has been manipulated). Due to the absence of globally accepted standards, blockchain suffers from a lack of interoperability. Many different factors, including as consensus models, transaction protocols, Cryptocurrencies, and smart contract capabilities, distinguish the many existing blockchain networks.

The researcher in [19] introduced a new architecture for a copyright management system that relies on digital watermarking and its associated data, such blockchain, by combining digital watermarking with the perceptual hash function, fast response code, and the Inter Planetary File System (IPFS). A peer-to-peer network may be used to centralize and finalize copyright management and the dissemination of copyrighted content without the need for an impartial third party. Cryptography is used by nodes to verify each other's identities and protect data. The danger of data loss, system failure, and other problems associated with relying on a centralized system is mitigated. However, no smart contract technology exists.

The primary objective of the work presented in [20] was to design, build, and evaluate a proof-of-concept prototype for preserving data integrity during video recording on an Android smartphone. The prototype also includes a web-based verification mechanism for checking the legitimacy of the recorded video. In order to provide an unchangeable record of transactions, blockchain technology stores information in a linear sequence of timestamps. A video's contents may be hashed cryptographically and then sent to a blockchain using an Android smartphone. Web-based clients perform video deconstruction in order to generate hashes that may be compared to those stored in the blockchain. Some of the required features are provided by a prototype system. The prototype, however, is restricted in that it cannot sign the generated hashes. Not only has it been hampered by the fact that it does not use HTTPS for communication, but the verification procedure also has to be streamlined before it can be used in practical settings.

In [3], the authors developed and deployed an innovative system for protecting against piracy of video content using blockchain and double watermarking. To extract keyframes from videos, they used the image correlation coefficient technique. They also work along with the contourlet transform domain and the sift algorithm to make invisible watermarks more secure against geometric assaults. Additionally, the authors precisely pinpointed the tamper site of the attacked video and finished the integrity verification of the watermarked video based on the features of the fragile watermarking. In light of the blockchain system's inherent inefficiency, the multimode server will be used to boost the effectiveness of the video copyright protection infrastructure.

Using non-subsampled contourlet transform (NSCT), singular value decomposition (SVD), and the blockchain, the authors in [21] suggested an innovative zero-watermarking method for video copyright. They investigated the potential and justifications for integrating zero-watermarking with blockchain technology. NSCT-SVD, the Arnold Transform, and a key-frames extraction method based on distance threshold form the basis of the zero-water-marking technique that was developed. Compared to DWT-SVD-based or older non-blocking features, the derived video features are more reliable and safe. The blockchain system optimized for video copyright practices performs admirably.

In [22], the authors suggested an improved method of encrypted watermarking that detects manipulation of robust video evidence. In a blockchain, each new block is cryptographically linked to the one before it in the network. The blockchain can be proven to be changed if any intermediate blocks are absent. The suggested task begins by dividing the video into individual frames based on the frame rate and length of the video. The selected watermark (genesis watermark) is combined with the first frame acquired using a series of image processing methods, including Logistic mapping, Integer Wavelet Transform (IWT), and singular value decomposition (SVD). The second frame is watermarked using the first frame's encoding. The blockchain-like structure in the verified video is the product of this process being applied to each successive frame.

There are a few problems with these earlier attempts. To begin with, the memory requirements of the solution are high because of the need to save a large number of coordinates that scale with the watermark's size (copyright). This is necessary to guarantee that the video's rightful owner can successfully extract the watermark from the watermarked video sequence. Second, attacks on the video frame texture region were possible with this method. Most of the above techniques embed the same watermark in all key frames, which leads to poor imperceptibility and resilience to video frame dropping, averaging, and swapping. New approaches increase resilience against video attacks by embedding various watermarks in all key frames according to different video scenarios. The watermark's invisibility is still poor, however.

Our approach involves real-time key-hashing of video content and chained storage of that content in chronological order, producing an inarguable repository. During verification, the same method is used on the video clip in order to obtain a hash value that can be compared to the hash stored on the blockchain. The main difference between the proposed approach and other approaches that rely on storing content-based video hashes in the blockchain is how to construct a perceptual hash function that represents information about intellectual property rights. The proposed method utilizes a video key frame extraction technique to build a perceptual hash function that minimizes the number of video frames needed to represent the required amount of video data while simultaneously cutting down on redundant frames and the number of calculations required to process the video data. The reader can find more recent works by looking at the Refs. [23–25].

## III. The proposed framework

In this paper, we propose a new design scheme for a digital watermark-based copyright management system, which combines digital watermarking, blockchain, and perceptual hash functions. The blockchain is used to safely store watermark information and to provide timestamp authentication for watermark data, thereby confirming the creation owner. To increase the reliability and storage capacity of digital watermarking, a perceptual hash function is employed to compute a hash value based on the structural information of video frames.

Keeping video files on the blockchain is impractical at this time. Hashing the video's frames, recording the key hash values of these frames in the blockchain, and storing the video files elsewhere for access is a more useful and handy approach. Traditional cryptographic hash methods, such as MD5 and SHA256, are not very appropriate for multimedia data such as video files. The computed results by conventional hash functions will be drastically changed due to video frame' attacks such as frame average attacks, frame deletion attacks, and frame reassembly attacks on the content structure. This is obviously not the desired outcome, so a novel hash method that is both resistant to and adaptable to content modification (normal operations and tempering) is required [26–28].

Before computing hash values, the perceptual hash function applies a sequence of processes to video frames, retaining only their structure data. Video hashing methods may be able to take advantage of the temporal property between successive frames, which is not available in image perceptual hashing methods. There are two basic categories of methods: those that focus on the spatial domain and those that combine the temporal and spatial domains. In most cases, the former chooses or creates the video's representative frames. A collection of hashes representing key or representative frames is what makes up the final hash. Perceptual video hashing techniques that use temporal-spatial domains produce hashes from data in both intra- and inter-frames [26,29–32].

### 3.1: Embedding phase

The scheme's components may be loosely broken down into two parts. The first part of the digital watermarking system covers the production of perceptual hashes, blockchain storage, the creation of digital watermarks for videos, and the embedding of digital watermarks. The second part discusses copyright data and the storage and dissemination of watermarked videos.

When considering blockchain's performance and security in the context of video watermarking applications, several assumptions are made, including: (1) Blockchain networks can face scalability issues when handling large volumes of data, such as video files and watermarks. (2) Blockchain's inherent design, which relies on consensus mechanisms, can introduce latency in transaction processing. (3) The transaction throughput of public blockchains is generally lower compared to centralized databases. (4) Storing large video files directly on the blockchain is impractical due to storage and cost constraints. (5) Blockchain networks are vulnerable to 51% attacks where an entity controls the majority of the network's hash power or stake. To implement blockchain effectively in video watermarking, these assumptions guide several strategic decisions, including utilizing off-chain storage that stores large video files on decentralized storage solutions like IPFS while storing only hashes and metadata on the blockchain to ensure scalability and efficiency. Furthermore, using Solidity and Truffle, blockchain-based video watermarking systems can effectively address some of the inherent weaknesses of blockchain technology, providing a robust solution for protecting digital video content. Using Solidity, a smart contract is created to store the hash, ownership details, and licensing information on the blockchain. Truffle is used to deploy and manage this contract.

Using Truffle, Solidity, and IPFS within the suggested public blockchain framework provides a robust solution for video watermarking, ensuring transparency, security, and decentralized storage. This approach leverages the strengths of public blockchains to create immutable records of video ownership and authenticity while using IPFS for efficient and decentralized storage. IPFS allows for decentralized storage of large video files, which would be impractical to store directly on the blockchain. By storing the video on IPFS and recording the hash on the blockchain, you can ensure both decentralization and efficient access.

Our choice of public blockchain stems from the fact that public blockchains are decentralized and maintained by multiple nodes, making it nearly impossible to alter data once it's recorded. This immutability ensures that the ownership and authenticity of a watermarked video can be reliably verified by anyone, fostering trust among content creators, distributors, and consumers. Furthermore, public blockchains are maintained by a decentralized network of nodes, reducing the risk of data being controlled or manipulated by a single entity. This decentralization can enhance security and resilience against attacks or failures; this openness increases accountability and reduces disputes over content ownership.

## Step 1: Video Splitting into Chunks

In this phase, we are going to extract individual frames from a video file with the help of Xconvert software (https://www.xconvert.com/) that breaks the data into fixed-size chunks or blocks from the beginning of the video file using the Fixed Size Chunking (FSC) procedure [33]. Algorithm 1 lists the main FSC procedure's main steps. In each chunk, key frames will be extracted according to the minimum amount of movement between frames.

```
Algorithm 1: Fixed Size Chunking (FSC) Procedure.
Input: Video File f, Integer chunk_ size
Output: Chunk List CL of chunks
Begin
List CL: = empty;
  Offset:  =  0;
            Chunk=GetChunk (f, offset, chunk_size);
            While chunk_size > 0 do
                          CL.add (chunk);
                          Offset: = Offset+ chunk_size;
                          Chunk=GetChunk (f, offset, chunk_size);
            End while
            Return CL;
 End
```

## Step 2: *Video Key Frames Extraction*

To speed up video browsing, summarization, indexing, and retrieval, video key frame extraction uses a small number of key frames to represent a large quantity of video content with minimal redundant frames and computational overhead [34,35]. Although while there are a number of widely-used key frame extraction techniques at present, most of them only work for specific videos and cannot be applied to other videos, and the key frames that are extracted are not necessarily indicative of the video's primary content [35–37]. Overall, the wide range of dance movements and the presence of too many redundant movements have restricted key frame extraction.

It may be necessary to extract one or more key frames from the shot, depending on its complexity. Because a shot is composed of images that are both continuous in time and substantively relevant, the most representative frames may be selected to serve as the shot's key frames

in order to include the most information [38–42]. Once the framing is complete, the suggested model estimates the optical flow of the frame image sequence of the movement's action video in order to select a group of key frames with reduced redundancy and summaries the video content. This technique can estimate optical flow for smaller objects by matching movements with large displacements. At the moment, the visual characteristics and content of the image frame do not yet vary significantly [35,43].

In this case, optical flow directional histogram features are utilized to explain the motion information of frame movements and describes the local appearance and shape features of movements.

$$
\begin{aligned}
E(w) = E_{\mathrm{color}}(w) + \gamma E_{\mathrm{grad}}(w) + \alpha E_{\mathrm{smooth}}(w) \\
+ \beta E_{\mathrm{match}}(w, w_l) + E_{\mathrm{desc}}(w_1)
\end{aligned}
\tag{1}
$$

$$
E_{\mathrm{color}}(w) = \int_\Omega \psi(|\nabla I_2(x + w(x)) - \nabla I_1(x)|^2)\mathrm{d}x
\tag{2}
$$

$$
E_{\mathrm{grad}}(w) = \int_\Omega \psi(|\nabla I_2(x + w(x)) - \nabla I_1(x))^2|\mathrm{d}x
\tag{3}
$$

$$
E_{\mathrm{smooth}}(w) = \int_\Omega \psi(|\nabla \mu(x)|^2 + |\nabla \nu(x)^2|)\mathrm{d}x
\tag{4}
$$

$$
E_{\mathrm{match}}(w) = \int_\Omega \delta(x)\rho(x)\psi(|w(x) - w_1(x)|^2)\mathrm{d}x
\tag{5}
$$

$$
E_{desc}(w_1) = \int \delta(x)|f_2(x + w_1(x) - f_1(x))|^2)dx
\tag{6}
$$

Let, the first and second frames to be aligned are represented by $I_1, I_2 : (\Omega \subset \mathbb{R}^2) \to \mathbb{R}^d$, where $d = 1$ for gray scale images and $d = 3$ for colour ones. In addition, the point $X := (x,y)^{\mathrm{T}}$ is in the image domain $\Omega$, and the optical flow field, which is a function $w : \Omega \to \mathbb{R}^2$, is denoted by $w := (u,v)^{\mathrm{T}}$. The correspondence vectors acquired via descriptor matching at certain positions $x$ are denoted by $w_1(x)$, and the robust function $\psi(s^2) = \sqrt{s^2 + \epsilon^2}$, where $\epsilon = 0.001$, is used to handle occlusions and other non-Gaussian deviations of the matching criteria. If a descriptor is accessible at position x in frame 1, then $\delta_i(x)$ (x) is 1, and otherwise, it is 0. There is a weight applied to each correspondence based on its matching score $\rho_i(x)$ (x). In frame 1 and frame 2, the (sparse) fields of feature vectors are represented by $f_1(x)$ and $f_2(x)$, respectively. The tunable weight parameters are α, ß, and γ, and the brightness invariance assumption $E_{\mathrm{color}}(w)$ is applicable to both colour and grayscale. Light will always have an impact. In order to reduce the impact of light, it is required to apply gradient constraint $E_{\mathrm{grad}}(w)$ to this base and then smooth it using $E_{\mathrm{smooth}}(w)$. The last two steps are to create descriptor matching and determine its minimal value using variable models and optimizations. For further information, see [35,38–42].

## Step 3: *Perceptual Hash Function*

In order to enable malicious-free operations, perceptual hashing for video is an efficient method for providing authentication of perceptual content. It identifies the video with a concise string and differentiates between secure content and attacked content [34]. Video hashing techniques could take into account the temporal property between successive frames, as opposed to image perceptual hashing techniques. Techniques may be broken down into two

categories: those that are based on the spatial domain and those that are based on both the spatial and temporal domains. In many cases, the former is responsible for choosing the video's representative or key frames. Combining the hashes of the key and representative frames yields the final hash. Perceptual video hashing techniques that are based on the temporal-spatial domain produce hashes using data from both inter- and intra-frames. Because to its robustness and sensitivity, perceptual video hashing has gained widespread acceptance. Sensitivity should be given preference when using video hashing for content authentication [35]. For simplicity, the suggested model follows the first category to extract perceptual hash function of the video's frames in which, video hash is made up of representative frames' hash. The hash of a video is the result of adding the hashes of each key frame that was obtained in the preceding phase. Note that the hash code length varies depending on the video's number of representative frames. Each representative frame, however, has the same length.

Let $V$ be a video sequence consisting of $n$ key frames denoted as $I_1, I_2, \ldots, I_n$. A perceptual video hashing function $H(V)$ maps the video sequence $V$ to a compact binary hash code or feature vector $h$ such that $h = H(V)$. The objective of perceptual video hashing is to ensure that the generated hash code possesses the following properties: (1) Robustness: The hash code should remain relatively unchanged under common transformations and distortions such as noise, compression, rotation, scaling, cropping, etc. (2) Discriminative: Similar videos should produce similar hash codes, while dissimilar videos should produce distinct hash codes. This property enables efficient retrieval and identification of similar video content. (3) Perceptual Meaningfulness: The hash code should capture perceptually meaningful information from the video content. In other words, perceptually similar videos should have hash codes that are close in Hamming distance. See [34,35] for more details.

Herein, the phash algorithm, an open-source perceptual hash library (https://www.phash.org), is employed to build the hash vector of each key frame. By superimposing the discrete cosine transform (DCT), which converts data from the spatial domain to the frequency domain, the pHash method is able to perform hashing. The algorithm keeps only the low-frequency coefficients of the DCT, typically the top-left 8x8 coefficients, and then computes the average value $m$ of the block. It compares the 64 DCT coefficients with the mean value and places 1 if the intensity is greater, 0 otherwise.

$$h_i = \{\{0, X_k < m\}, \{1, X_k \geq m\}, \forall k \in [0, 63]\} \tag{7}$$

The resulting hash value for each key frame is a binary vector of fixed length, typically 64 bits for pHash. To cut down on repetition, the hash values of the frames are compared to find ones that seem identical. The remaining frames are picked as key frames, the key frames that contain the most important content of the video chunk.

Using perceptual hash functions for video watermarking combined with blockchain technology can significantly enhance content verification and integrity. Addressing scalability challenges involves efficient hashing algorithms, segment-based hashing, and off-chain storage solutions like IPFS. In our suggested model, we utilize a simplified version of pHash that can reduce computational overhead to generate compact hash values quickly. Furthermore, instead of generating perceptual hashes for every frame, divide the video into segments (e.g., every few seconds or key frames) and generate a hash for each segment. This reduces the number of hashes while still providing adequate verification. Finally, using IPFS to store the perceptual hashes and video files off-chain using decentralized storage solutions like IPFS. Only store the IPFS hash (a unique identifier) on the blockchain, which points to the actual data stored off-chain. By implementing these strategies, it is possible to achieve a scalable and robust system for managing and verifying video content in a decentralized manner.

The computational complexity of building perceptual hash functions for video watermarking is an important factor to consider, especially for scalability and real-time applications. This process generally depends on the specific algorithm used and the size of the video content. Its complexity is $O(n\log n)$ for the DCT computation, where $n$ is the number of pixels in the resized image/frame. So the total complexity is $O(F \times H)$, where $F$ is the number of frames and $H$ is the complexity of the perceptual hash function.

### Step 4: *Watermark Embeddings*

This step describes the process of embedding a watermark into the keyframes, which involves the following phases [43]:

- Convert keyframe ($K_f$) into *YUV* color space and select the chrominance component ($Kv$).

- The chrominance component ($Kv$) is decomposed into four sub-bands by performing a 1-level lifting wavelet transform (LWT) and considering sub-band ($LL_v$) for embedding purpose.

- Perform Hessenberg transform (HT) on sub-band $LL_v$ for reducing a matrix to tridiagonal form

$$HT[LL_v] = Q_v \times H \times Q_v^T \tag{8}$$

$Q_v$ is the orthogonal matrix that is used to perform HT and IHT (inverse HT), $H$ is a Hessenberg matrix that is obtained by applying HT on $LL_v$, and $QT_v$ is the transpose of $Q_v$.

- Select squared watermark logo ($w$) and perform Arnold transform to obtain scrambled watermark $S_w$ for security enhancement purpose.

- First, a 1-level LWT is used to break the watermark $S_w$ into four sub- bands. Then, the sub-band $HL_w$ is chosen for further processing.

- Perform HT on sub-band $LL_w$

$$HT[LL_w] = Q_w \times H \times Q_w^T \tag{9}$$

- Apply intelligent gravitational search algorithm (IGSA) to obtain the set of multiple scaling factor α. See [43] for more details.

- Embed the component $\underline{H_w}$ into $H_v$

$$\tilde{H}_v = H_v + \alpha * H_w \tag{10}$$

- Apply inverse HT (IHT) on modified matrix ($\tilde{H}_v$) to obtain a Hessenberg embedded matrix $\tilde{LL}_v$

$$\tilde{LL}_v = Q_v * \tilde{H}_v * Q_v^T \tag{11}$$

- Apply inverse LWT (ILWT) on matrix ($\tilde{LL}_v$) to obtain the watermarked video frames ($W_v$):

- In order to acquire the watermarked video ($V^*$), it is necessary to transform all *YUV* watermarked video keyframes into RGB watermarked video keyframes and then combine them with the remaining video frames.

### Step 4: *Video Uploading using IPFS Blockchain*

With this module, we can use Inter-planetary File Systems (IPFS), a peer-to-peer distributed file system (see Fig 2) that seeks to replace HTTP and build a better web accessing, to upload video files (watermarked version) with their details and share them in exchange for credit. The submitted information includes the copyright owner's name, email address, and the file's title [33–35]. When a file is uploaded to IPFS, watermarked video $V^*$, it is broken up into smaller pieces called "chunks," with a limit of 256 kilobytes per chunk for data and/or links to other chunks. Each chunk is identified by its own cryptographic hash, also known as a content identifier, calculated from its content.

There are various issues with the conventional centralized storage model, such as the need for many large-scale server storage devices. This raises operational costs, and if the server loses power due to physical damage or other major concerns, it will have a negative impact on the convenience and utility of video users. What's worse is that if hackers manage to breach the server, it will either leak or delete a lot of crucial data, leading to massive and irreversible losses [35]. IPFS is a decentralized file sharing, networking, and content distribution system. In contrast to HTTP, IPFS doesn't care where a file is physically stored and ignores the file's path and name. It only cares about what is really in the file, not what may be there [36].

A cryptographic hash is computed from the contents of a file once it is uploaded to an IPFS node. In order to verify a file's integrity, IPFS employs a distributed hash table to determine

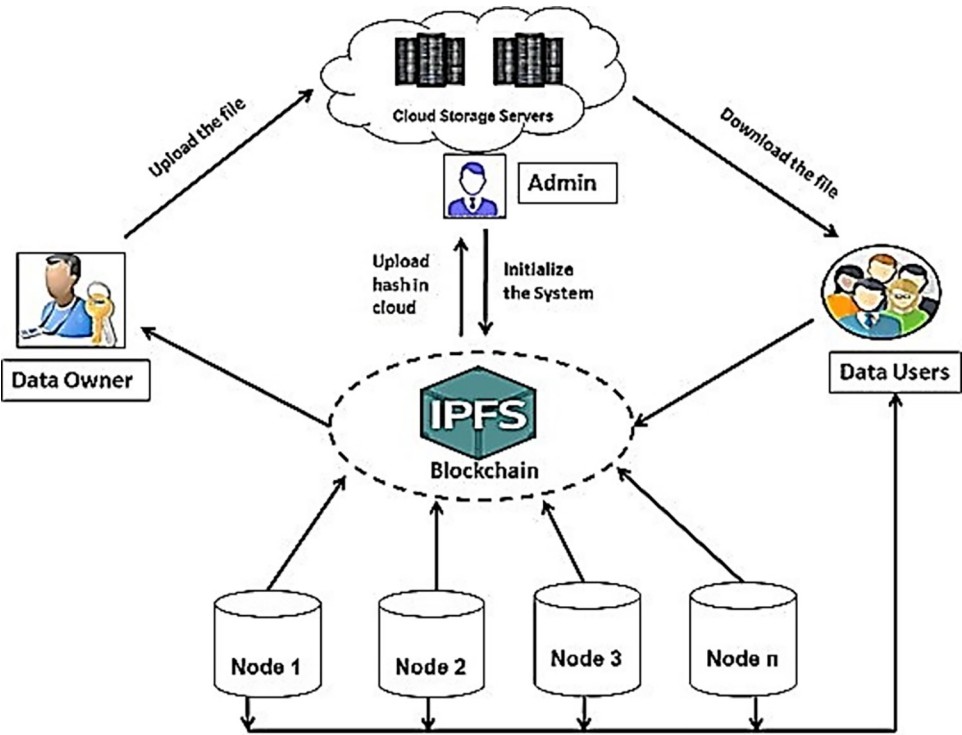

**Fig 2. Hierarchical structure and IPFS based semi-decentralized system.**

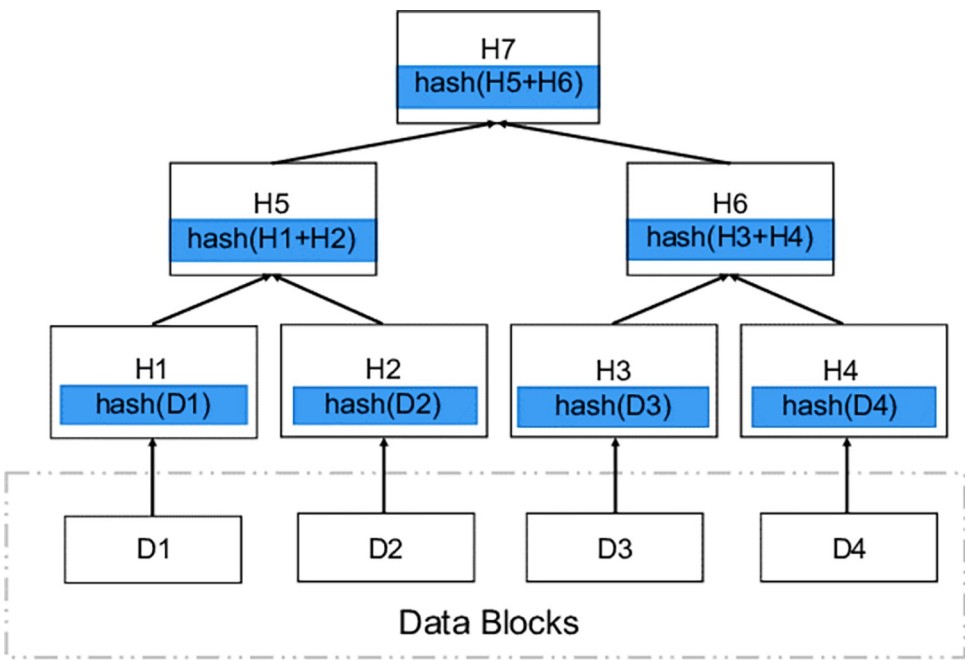

**Fig 3. An Illustration of Merkle DAGs.**

the node where the file is stored, then gets the file from that node. As a result, storing videos on IPFS may increase their safety while decreasing the platform's operational costs. In addition, IPFS has an HTTP gateway, so that anybody with an internet connection and a browser may access and download videos.

IPFS is built on top of technologies like Directed Acyclic Graphs (DAGs) and Distributed Hash Tables (DHTs) to enable content storage with content linking and content addressing. Merkle DAGs are a particular kind of DAG used by IPFS that are a kind of self-verifying data structures. When a node's contents are hashed using the node's unique payload and the list of content it presently contains, a Merkle Directed Acyclic Graph is created. The content identifier CID of an IPFS node is permanently connected to the contents of that node's payload and all of that node's descendants (see Fig 3). A CID is a label that is used to refer to content in IPFS. While it does not specify where the content is kept, it does create a kind of location based on the content itself. Regardless of the size of their underlying content, CIDs are short. The cryptographic hash of the content serves as the foundation for CIDs.

When a file is uploaded to IPFS, it is immediately split into 256kB parts and a Merkle DAG is generated. The root node takes the distinct CIDs assigned to each chunk and merges them into a single CID called the root CID. Merkle directed acyclic graphs (DAGs) are special because they enable chunk de-duplication by letting nodes have many parent nodes. To preserve storage and bandwidth resources on the network, de-duplication is used when identical content is kept on the network instead of being transported multiple times. For further information on IPFS, refer to [44–49].

In the context of IPFS, Bitswap is a crucial component responsible for efficiently exchanging blocks of data between peers in the network. Here's a detailed overview of how Bitswap operates within IPFS [50]:

- Block Retrieval: When a node in the IPFS network wants to retrieve a file or a specific block of data, it sends out a request to its connected peers.

- Wantlist: Along with the request, the node sends a "wantlist" to its peers. This wantlist contains entries for each block it currently desires but does not possess. Each entry typically includes the CID (Content IDentifier) of the desired block and the corresponding priority.

- Responses: Peers that receive the request check their own local storage to see if they have any of the requested blocks. If they do, they respond with those blocks.

- Block Prioritization: Upon receiving a request along with a wantlist, peers prioritize which blocks to send based on various factors including the urgency indicated in the wantlist and their own strategies for optimizing data exchange.

- Block Trading: After receiving blocks from peers, nodes may redistribute these blocks to other peers in need. This establishes a sort of block trading system where nodes help each other acquire the data they need.

- Duplicate Block Prevention: Bitswap includes mechanisms to prevent duplicate blocks from being sent unnecessarily. Peers keep track of which blocks they have already sent to specific peers, reducing redundant data transmission.

- Data Verification: Both sending and receiving nodes verify the integrity of the transmitted blocks using cryptographic hashes. This ensures that the received data hasn't been tampered with during transmission.

- Request Manager: Bitswap also includes a request manager component responsible for coordinating block requests and managing peer connections efficiently.

- Connection Management: Bitswap dynamically manages connections with peers in the network, establishing new connections as needed and optimizing the utilization of available network resources.

In general, the integration of video watermarking with blockchain offers a secure and tamper-proof solution for copyright protection. However, a major concern arises with scalability, particularly when dealing with large volumes of video content. The proposed method effectively addresses these challenges through IPFS storage and Merkle DAGs. Instead of storing the entire watermarked video directly on the blockchain, the system utilizes IPFS for efficient storage and retrieval of video data. IPFS's distributed storage architecture and content-based addressing mechanisms provide a scalable and cost-effective solution for handling large video files (off-chain watermark storage)".

Furthermore, to maintain data integrity while reducing blockchain storage requirements, the proposed method employs Merkle DAGs (directed acyclic graphs) to store hashes of the watermarked video content on the blockchain. Merkle DAGs allow for more efficient updates and verification of data compared to traditional Merkle trees. This is particularly beneficial for video watermarking, where new watermarks may be added or existing ones updated frequently. Additionally, Merkle DAGs are immutable, meaning once a hash is added, it cannot be altered. This ensures the integrity of the watermark data and prevents unauthorized modifications. Finally, Merkle DAGs provide greater flexibility in organizing and structuring data, making them well-suited for handling complex data relationships like those encountered in video watermarking.

## 3.2: Verification phase

After the embedding process is complete, the video file with the watermark and its associated block information may be uploaded to the IPFS network. Next, access the IPFS, find the video with the watermark, and download it together with the copyright information. Meta data,

including the perceptual hash value and the information about the copyright owner, serves as a digital signature for the copyright owner. These, together with the cryptographic hash value, are stored in the blockchain and used as supplementary transaction information to start a request for a transaction on the network.

In addition, there is a step for the user to verify the video copyright after obtaining the video, as shown below:

1. *Upload the Watermarked Video to IPFS*: Securely store video items using the IPFS framework and generate an addressable hash of the uploaded chunked video with watermarks.

2. *Calculate the Video Hash*: Perceptual hashes are designed to measure the perceived similarity between two media assets, while cryptographic hashes are intended to ensure the authenticity and integrity of data. When uploading perceptual hashes to IPFS, the platform generates a cryptographic hash value for the uploaded data using its integrated hash function. Herein, the same procedure as in step 3 in the embedding phase was used to calculate both type of hashes.

Herein, we use the Truffle framework to set up a system structure and run Solidity code to generate blocks and hash values. By using Ethereum, the Truffle framework facilitates the development of decentralized applications on the blockchain. Truffle provides a suite of tools that promotes the use of the solidity programming language for the creation of smart contracts. On top of that, it provides the infrastructure for evaluating smart contracts and the resources to implement them in blockchain network technology for commercial transactions.

Solidity is a statically-typed programming language designed for developing smart contracts that run on the Ethereum Virtual Machine (EVM). Smart contracts are self-executing contracts with the terms of the agreement directly written into code. For blockchain-based video watermarking, Solidity can be used to develop smart contracts that manage various aspects of watermarking, including: (1) Ownership Verification: Smart contracts can be designed to store and verify the ownership of video content. When a video is uploaded, its unique watermark can be registered on the blockchain along with the owner's information. (2) Watermark Embedding and Extraction: While the actual embedding and extraction of watermarks are performed off-chain, smart contracts can be used to log these actions on the blockchain. For example, a hash of the watermarked video can be stored on-chain to verify its authenticity later. (3) Usage Tracking: Smart contracts can track how and when the video is accessed or used. Each time the video is played or shared, a record can be added to the blockchain, ensuring a transparent and immutable history of the video's usage. (4) License Management: Smart contracts can manage the issuance and verification of licenses for video content. This ensures that only authorized users can access or distribute the video, reducing the risk of piracy.

Truffle is a development framework for Ethereum that provides tools to make writing and managing smart contracts easier. It includes features for compiling, deploying, and testing contracts, which are crucial for ensuring the reliability and security of blockchain applications. In the context of video watermarking, Truffle can help in the following ways: (1) Smart Contract Development: Truffle provides a structured environment for developing Solidity smart contracts. It includes libraries that simplify common tasks, such as interacting with the blockchain and managing contract state. (2) Automated Testing: Truffle allows for the creation of automated tests to ensure that smart contracts behave as expected. This is essential for verifying that the watermarking process is secure and that the contracts correctly handle ownership verification, usage tracking, and license management. (3) Deployment Management: Truffle simplifies the deployment of smart contracts to different blockchain networks. This includes

managing migrations and ensuring that contracts are deployed in a consistent and reproducible manner. (4) Interaction with Frontend: Truffle integrates well with frontend technologies, enabling the development of user interfaces that interact with smart contracts. This is important for creating user-friendly applications for video content creators and consumers.

3. *Check the calculated Hash against the Stored Hash*: The calculated hash value and the hash value in the blockchain are compared with each other to determine the copyright. The hamming distance was used for comparison. Hamming distance between two binary strings of equal length is the number of positions at which the corresponding bits are different. Mathematically, it can be expressed as:

$$dH(x, y) = \sum_{i=1}^{n} (x_i \oplus y_i) \tag{12}$$

$dH(x,y)$ is the Hamming distance between binary strings $x$ and $y$, $n$ is the length of the binary strings $x$ and $y$, $x_i$ and $y_i$ are the individual bits at position $i$ in the binary strings $x$ and $y$, respectively, $\oplus$ is the bitwise XOR operation, which compares corresponding bits in two binary strings and returns 1 if they are different, and 0 if they are the same. If the similarity between the hashes is less than 100%, it suggests a potential violation of copyright. At this point, a complete process has ended.

The use of blockchain for video watermarking in a copyright protection system can significantly influence both performance and responsiveness. Here are the key implications: (1) as more videos are watermarked and registered on the blockchain, the size of the blockchain grows. This can affect the speed of transaction validation and block creation. (2) Blockchain transactions, especially in public blockchains, can have a high latency due to the time required for block confirmation. This can delay the process of watermark registration and verification. (3) blockchain's decentralized nature means that every transaction needs to be confirmed by the network, which can take time. This can affect the ability to perform real-time watermark verification. (4) the responsiveness of the system can vary based on the number and distribution of nodes. More nodes can mean more security but also potentially more latency.

By effectively utilizing IPFS and Merkle DAGs, the proposed method demonstrates a well-considered approach to addressing scalability challenges in blockchain-based video watermarking. The strategies employed were to minimize storage overhead, improve transaction throughput, and optimize performance for handling large volumes of video content. The use of Merkle DAGs in particular provides additional advantages in terms of immutability, efficiency, and flexibility, making it a suitable choice for securing and managing video watermark data in a scalable and secure manner.

In summary, there are other areas outside of video forensics that might benefit from using video watermarking solutions that are based on the blockchain and focus on digital media copyright enforcement, including (1) Enhanced Copyright Protection: Blockchain's immutable ledger can store detailed ownership information. Once a video is watermarked and registered on the blockchain, the ownership record cannot be altered or deleted, providing a clear and permanent record of who owns the rights to the content. (2) Streamlined Licensing and Royalties: Blockchain can automate licensing agreements through smart contracts. These self-executing contracts ensure that creators are automatically compensated when their content is used, reducing the complexity and cost associated with traditional licensing processes. (3) Traceability: Watermarked videos can be traced back to the source if pirated copies appear online. Blockchain can record every transaction and access point, making it easier to identify and take action against unauthorized distribution.

## IV. Experimental results

In this section, we assess the suggested model by conducting several experiments that are simulated on MATLAB 2020a on an Intel(R) Core (TM) i5-6200U CPU 2.4 GHz, 8 GGB of memory, and Windows OS. Eight sequences of videos benchmark dataset available from the website https://vision.middlebury.edu/stereo/ using public ground truth were subjected to the assessment. There are six datasets total; three of them, "Dimetrodon," "RubberWhale," and "Hydrangea," include real-world key frames with complicated occlusions, while the other four, "Grove2," "Grove3," "Urban2," and "Urban3," comprise synthetic computer-generated images. Venus, the last series, is made up of stereo frames [51].

The primary goal of the first set of experiments is to test how well optical flow techniques for video key frame extraction work and how accurate they are. Here we assess three distinct optical flow approaches and select the most appropriate descriptor type: one based on geometric blur (GB) [52], one based on histograms of oriented gradients (HOG) [51], and one based on region matching [53], where region correspondences are used to recover large displacements. The experiment also contrasts these techniques with one that just uses warping that aligns each subsequent frame with the reference frame and does not use descriptors.

The Average Angular Error (AAE) metric, which determines the average angle between the actual and estimated flow vectors, is used to assess the performance. The optical flow becomes more precise as this measure decreases. Table 1 shows that with an average AAE of 3.40, the warping technique produces the best results across all scenarios, followed by the HOG descriptor with an average AAE of 3.96. At an average AAE of 4.15, the region-matching approach produces the worst results across all scenes. There are a number of restrictions when it comes to digital video processing that use optical flow based on warping. The first thing to keep in mind is that the warping technique is effective up to the point when the relative motion of the small structures is greater than their own scale. However, this method becomes ineffective beyond this point. When this occurs, the large-scale structures' predictions of motion diverge significantly from the actual one.

HOG features are computed over local regions of an image (video frame), capturing localized motion patterns. This can be advantageous in optical flow estimation, where motion may vary across different parts of an image. By analyzing local motion patterns, HOG can help detect and estimate complex motion dynamics. Furthermore, HOG descriptors reduce the dimensionality of feature space by quantizing gradient orientations into bins. This reduction can help in speeding up computation and reducing memory requirements in optical flow estimation, especially in real-time applications. Finally, HOG descriptors are somewhat invariant to common image transformations such as small changes in scale and rotation. This property

**Table 1. A quantitative comparison of the optical follow descriptors used to extract key frames in terms of Average Angular Error (AAE) metric.**

| Datasets / Descriptor Methods | Warping | Region Matching | HOG | GB |
|---|---|---|---|---|
| Dimetrodon | 1.85 | 1.76 | 1.87 | 1.96 |
| Grove2 | 2.11 | 2.28 | 2.69 | 2.80 |
| Grove3 | 5.60 | 6.57 | 6.40 | 6.38 |
| Urban2 | 2.29 | 3.07 | 2.67 | 3.18 |
| Urban3 | 3.99 | 5.79 | 5.10 | 5.21 |
| RubberWhale | 3.80 | 3.87 | 3.97 | 4.18 |
| Hydrangea | 2.36 | 2.39 | 2.47 | 2.58 |
| Venus | 5.22 | 7.40 | 6.48 | 6.56 |
| **Average** | **3.40** | **4.15** | **3.96** | **4.10** |

**Table 2. Robustness performance of the suggested method for the videos under consideration when subjected to different types of attacks in terms of mean normalized correlation (MNC).**

| Attack index | Video attack | Silent | Foreman | Mobile | Hall monitor |
|---|---|---|---|---|---|
| MF | Median filtering (3 × 3) | 0.9996 | 0.9992 | 0.9998 | 0.9994 |
| WF | Wiener filtering (3 × 3) | 0.9999 | 0.9994 | 0.9997 | 0.9993 |
| GF | Gaussian filtering (3 × 3) | 0.9998 | 0.9995 | 0.9993 | 0.9997 |
| RO | Rotation (45˚) | 0.9997 | 0.9993 | 0.9999 | 0.9996 |
| TR | Translation (30, 30) | 0.9995 | 0.9997 | 0.9996 | 0.9992 |
| CR | Cropping CR (center) | 0.9997 | 0.9995 | 0.9993 | 0.9998 |
| SH | Sharpening | 0.9997 | 0.9996 | 0.9995 | 0.9991 |
| GC | Gamma correction (γ = 0.6) | 0.9992 | 0.9998 | 0.9997 | 0.9994 |
| HI | Histogram equalization | 0.9998 | 0.9993 | 0.9996 | 0.9999 |
| GN | Gaussian noise (0, 10%) | 0.9991 | 0.9995 | 0.9992 | 0.9997 |
| SP | Salt and pepper noise (10%) | 0.9997 | 0.9993 | 0.9995 | 0.9996 |
| PN | Poisson noise | 0.9994 | 0.9999 | 0.9997 | 0.9992 |

can be beneficial in optical flow estimation scenarios where the relative motion between objects might cause such transformations. So, by utilizing HOG descriptors, the loss in accuracy is a price worth paying for the ability to capture much larger displacements (relative to the warping approach, there is an average accuracy loss of 16% with AAE.). As a consequence, the suggested model follows the HOG descriptor to extract key frames.

The purpose of the second set of experiments is to demonstrate that the suggested technique for embedding watermarks is successful against various kinds of attacks. Four standard benchmark videos were used to assess the proposed approach and twelve video attacks were tested, with an emphasis on imperceptibility factor as illustrated in Table 2. The first row displays videos, while the leftmost column lists attacks. To determine robustness values, we used the mean normalized correlation (MNC), an essential metric for evaluating video watermarking methods. By comparing the embedded and extracted watermarks from the watermarked video frames, MNC is able to quantify the correlation. An improved watermarking approach is shown by a greater MNC value, which means that the watermark is recovered more effectively. Table 2 shows that under diverse attacks, the suggested method gets high values for MNC metric.

The proposed model's use of the wavelet transform for embedding might be a contributing factor to these results. By applying a one-level linear wavelet transform on the chrominance channel of the motion frames—and by choosing the low-frequency sub-band LL in particular—the watermark becomes more imperceptible and resilience to attacks. The perceived quality of the video is preserved by embedding the watermark in the chrominance component, which minimizes any visual distortions induced by the watermark.

In general, wavelet-based watermarking techniques can leverage perceptual masking properties, embedding the watermark in visually insignificant coefficients that are less likely to be affected by attacks without perceptible degradation in image quality. Furthermore, wavelet-based watermarking allows for adaptive embedding strategies, where the strength and location of the watermark can be adjusted based on image characteristics or anticipated attacks. Adaptive techniques improve robustness by tailoring the watermarking process to specific image properties or attack scenarios [54,55].

**Table 3. One-way ANOVA on different types of attacks applied to the videos.**

|  | Video | N | Mean | Std. deviation | F | P-value | Result |
|---|---|---|---|---|---|---|---|
| **Attacks** | **Silent** | 12 | 0.999592 | 0.00246 | 0.525 | 0.66 | N.Sig |
|  | **Foreman** | 12 | 0.999500 | 0.00022 |  |  |  |
|  | **Mobile** | 12 | 0.999567 | 0.00021 |  |  |  |
|  | **Hall monitor** | 12 | 0.999492 | 0.00026 |  |  |  |

ANOVA (Analysis of Variance) analysis can be a powerful tool for comparing the performance of video watermarking systems under different attack scenarios in terms of mean normalized correlation (MNC). In this case, we clearly define the experimental design, including: Independent Variable: Types of attacks (e.g., compression, noise addition, cropping). Dependent Variable: Mean Normalized Correlation (MNC). Control Variables: Factors that may influence MNC but are not the focus of comparison (e.g., watermarking algorithm, video content). After that, we collect MNC values for each attack scenario from multiple trials or samples ($N = 12$). We apply a one-way ANOVA to test for significant differences in MNC values among different attack scenarios.

- The null hypothesis (H0) is that there is no significant difference in the mean of MNC across attack types.

- The alternative hypothesis (H1) is that at least one attack type differs significantly in the mean of MNC.

In this case, the F-statistic indicates whether there are significant differences among group means, and the p-value indicates the probability of observing the data if the null hypothesis were true. The results of a one way ANOVA (F-test) are revealed in Table 3. The results confirm that (1) there are no statistically significant differences between the performances of the videos under different attacks, as the value of F was 0.525, with a significant level greater than 0.05. (2) This could be explained by the fact that all videos are resistant to different types of attacks.

Regarding the suggested model's resistance to more advanced manipulations, such as deepfake technology or sophisticated frame interpolation attack. Deepfake algorithms primarily focus on the luminance channel to ensure high visual fidelity and realism in terms of shapes and edges [56]. In our suggested model, embedding watermarks in the chrominance channel can therefore survive these manipulations better than those in the luminance channel. Deepfake algorithms might not accurately replicate the subtleties of the chrominance channels, watermarks embedded in these channels can help in detecting inconsistencies and manipulations. By comparing the extracted watermark from a potentially deepfaked video with the original, it is possible to identify tampering. Also, as our model embeds the watermark in the frequency domain; it can make the watermark more resilient to deepfake manipulations [57].

Frame interpolation attacks involve generating intermediate frames between existing frames in a video to create smooth transitions, often using techniques like deep learning-based algorithms. This type of manipulation can distort or disrupt embedded watermarks. Our strategies for enhancing watermark robustness against frame interpolation rely on distributing the watermark information across multiple frames instead of embedding it in single frames. This way, even if some frames are interpolated or modified, the watermark can still be reconstructed from the remaining frames. Furthermore, adjust the watermark embedding process based on the content of each frame. This approach can use characteristics like motion vectors

or scene changes to decide where and how to embed the watermark, making it more resilient to interpolation artifacts [58–60].

Handling a 51% attack on a blockchain used for video watermarking involves several strategies to mitigate the impact and prevent the attack. A 51% attack occurs when a single entity or a group of entities gains control of more than 50% of the network's mining or hashing power, allowing them to manipulate the blockchain. Here are some strategies used within the suggested model to handle and prevent such attacks within the context of video watermarking on a blockchain: (1) our blockchain network structure has a large and diverse number of nodes. The more nodes there are, the harder it is for any single entity to control the majority. (2) In our model, by using IPFS, the actual video content and watermarks are stored in a decentralized manner, making it difficult for any single entity to tamper with or control the data. Once data is stored on IPFS, it is immutable and can be verified using cryptographic hashes. While the perceptual hash is stored on the blockchain, the actual video data is stored on IPFS, which is a decentralized storage network. This reduces the impact of a 51% attack on the overall system. (3) The perceptual hash of a video (a compact representation of the video's content) is small in size. Storing this hash on the blockchain instead of the entire video reduces the amount of data that needs to be processed and stored on-chain.

Testing how well the suggested model for video copyright protection works with IPFS and blockchain is the major goal of the third set of experiments. In terms of execution time, the results of IPFS main procedure are described. As part of our model's testing, we used an IPFS upload operation with 70 peers, including photographers, graphic designers, and video providers. In Table 4, we can see the different numbers of peers and the calculation time for files of different sizes that are uploaded into the IPFS secure distributed storage layer. There is a clear correlation between the file size and the computing time. The time required to calculate the upload takes a hit as the number of peers grows.

In general, for computer resources, processing and transferring larger files need more processing power and memory. How well a computer handles bigger files is dependent on its processing capability, which includes factors like CPU speed and memory capacity. For network resources, more data must be sent across the network to transfer bigger files. When it comes to file transfers, latency and bandwidth are two of the most important factors. The processing time for bigger files may be affected by factors such as network congestion or limited bandwidth. Optimizing these resources can help reduce computation time and improve the efficiency of handling larger files in the suggested model.

The proposed blockchain–based technique has been compared with traditional two recent non-blockchain video-watermarking techniques presented in Refs. [61,62]. In Ref.[61], a lossless video watermarking system is suggested that operates by selecting keyframes optimally via the use of a linear wavelet transform and an intelligent gravitational search algorithm. Using the histogram difference approach, this methodology extracts color motion as well as static

**Table 4. Execution time analysis for uploading files on IPFS off-chain storage with varying number of peers.**

| Number of Peers | 4MB Execution Time (s) | 8MB Execution Time (s) | 16MB Execution Time (s) | 32MB Execution Time (s) |
| --- | --- | --- | --- | --- |
| 10 Peers | 0.6 | 1.1 | 1.8 | 3.3 |
| 20 Peers | 0.8 | 1.4 | 2.2 | 3.6 |
| 30 Peers | 1.1 | 1.8 | 2.6 | 3.8 |
| 40 Peers | 1.3 | 2.1 | 2.9 | 3.9 |
| 50 Peers | 1.4 | 2.2 | 3.1 | 4.2 |
| 60 Peers | 1.7 | 2.5 | 3.3 | 4.4 |
| 70 Peers | 1.9 | 2.8 | 3.4 | 4.7 |

**Table 5. The imperceptibility (MPSNR, MSSIM) of the considered techniques under different attacks.**

| Cover video | Our blockchain-based Technique | | Non-blockchain-watermarking Technique in Ref. [61] | | Non-blockchain-watermarking Technique in Ref. [62] | |
|---|---|---|---|---|---|---|
| | MPSNR | MSSIM | MPSNR | MSSIM | MPSNR | MSSIM |
| Silent | 48.98 | 0.9997 | 48.97 | 0.9995 | 42.65 | 0.9897 |
| Foreman | 48.05 | 0.9999 | 47.98 | 0.9997 | 42.57 | 0.9889 |
| Mobile | 49.34 | 0.9994 | 48.07 | 0.9993 | 41.98 | 0.9843 |
| Hall monitor | 48.87 | 0.9999 | 48.52 | 0.9999 | 42.37 | 0.9863 |
| **Average** | **48.89** | **0.9998** | **48.39** | **0.9996** | **42.39** | **0.9873** |

frames from the cover video. The chrominance channel of the motion frames undergoes a one-level linear wavelet transform, and a low-frequency sub-band LL is chosen for the watermark embedding. In Ref. [62], a secure video watermarking method is provided employing a novel two-dimensional complex map. In order to demonstrate the chaotic nature of the suggested map, standard analyzes have been done on a dynamical system. There is also an efficient approach for embedding and extraction watermark that uses single value decomposition in conjunction with the IWT, DWT, and CT transforms.

This set of experiments measures the imperceptibility performance of the proposed technique against existing techniques in terms of mean peak signal to-noise ratio (MPSNR) and Mean Structural Similarity Index Measure (MSSIM) under different video attacks. See [61] for more details regarding these metrics. Table 5 highlights the MPSNR and MSSIM values corresponding to each video against compared schemes. It can be noticed from the table that the proposed technique attains an average value of MPSNR as 48.89 and MSSIM value as 0.9998. These superior values confirm that the proposed technique outperforms the compared schemes. In general, Blockchain employs cryptographic techniques via hashing to ensure the security and privacy of data. Watermarking information stored on a blockchain can be encrypted and accessible only to authorized parties, reducing the risk of unauthorized access or manipulation. Furthermore, Table 6 compares the suggested blockchain-based video watermarking approach with non-blockchain approaches.

## V. Limitations and challenges

Integrating blockchain technology with digital watermarking and perceptual hash functions holds significant potential for enhancing content security, authenticity verification, and

**Table 6. Comparison between non-blockchain technique and blockchain-based video watermarking technique.**

| Aspect | Traditional Non-Blockchain technique | Blockchain-Based technique |
|---|---|---|
| Storage | Centralized | Decentralized via IPFS |
| Data Integrity | Managed by central authority | Ensured by blockchain immutability |
| Security | Vulnerable to tampering | High security with cryptographic methods |
| Tamper Resistance | Lower | Higher |
| Transparency | Limited | High due to public ledger |
| Audit Trail | Challenging | Clear and immutable |
| Robustness to Changes | Limited | High with perceptual hashing |
| Single Point of Failure | Yes | No |
| Scalability | Generally scalable | Potential issues with large-scale data |
| Verification | Manual and centralized | Automated and decentralized |

ownership tracking. However, this integration presents a range of limitations and challenges including:

- Digital watermarks and perceptual hash functions generate data that needs to be stored or referenced on the blockchain. Storing large amounts of data directly on the blockchain can be expensive and impractical due to limited storage capacity and high transaction fees.

- Blockchain networks, especially those with public and permissionless configurations, often face scalability issues. The processing speed of transactions might not be fast enough to handle real-time applications of digital watermarking.

- The additional computational steps required for watermark embedding, hash generation, and blockchain transactions can introduce latency, which might not be acceptable in real-time applications.

- Ensuring compliance with data privacy regulations is crucial when storing or referencing any personal or sensitive information on the blockchain. One potential solution is to store personal data off-chain and use the blockchain to store references or hashes of the data. This way, the actual data can be deleted or modified off-chain while maintaining the integrity and immutability of the blockchain records.

## VI. Conclusion

The objective of this work is to propose an enhanced video copyright protection model that integrates both watermarking and blockchain technologies into a unified framework able to handle watermarking-based copyright difficulties such as pose integrity after embedding watermarks. In the suggested model, watermarks are utilized to convey the copyright owner's personal information, whereas blockchain data structures are employed to ensure the integrity of the watermark data so that it will be hard to change. The proposed framework consists of important phases, which include uploading multimedia objects and cutting frames, embedding watermarks on frames, setting up a system structure, creating perceptual hashes, and matching objects.

Different sets of experiments were conducted to determine the impact of different types of attacks on the visibility and detectability of the embedded watermark, quantifying the degradation in watermark quality or retrieval accuracy, and the results confirm the robustness of the suggested model in terms of the MNC metric with an average of 99%. Regarding blockchain transaction throughput in terms of execution time of uploading files on IPFS off-chain storage with varying numbers of peers to assess the scalability and efficiency of the blockchain network for handling video copyright protection tasks, the results give reasonable throughput.

Here are some potential directions for further research and development for this work: (1) Investigate privacy-preserving techniques to protect sensitive information while still allowing for effective watermarking of video content. (2) Utilize smart contracts to automate the process of watermarking video content, including payment transactions, rights management, and royalty distribution. (3) Address the scalability challenges associated with blockchain technology to accommodate the large volume of transactions and data generated by video watermarking systems. (4) Currently, we haven't conducted a dedicated evaluation of the model's resistance against deepfakes. However, based on the watermark placement strategy, we believe the impact would be limited in most cases. Further investigation using deepfake datasets specifically designed to target watermarked content is necessary to fully assess the model's robustness in this scenario.

## Author Contributions

**Conceptualization:** Saad Mohamed Darwish.

**Data curation:** Mona Mahamod Abu-Deif.

**Formal analysis:** Saad Mohamed Darwish, Saleh Mesbah Elkaffas.

**Funding acquisition:** Mona Mahamod Abu-Deif.

**Investigation:** Mona Mahamod Abu-Deif, Saleh Mesbah Elkaffas.

**Methodology:** Mona Mahamod Abu-Deif.

**Project administration:** Mona Mahamod Abu-Deif.

**Resources:** Mona Mahamod Abu-Deif.

**Software:** Mona Mahamod Abu-Deif.

**Supervision:** Saad Mohamed Darwish, Saleh Mesbah Elkaffas.

**Validation:** Saad Mohamed Darwish.

**Writing – original draft:** Saad Mohamed Darwish, Saleh Mesbah Elkaffas.

**Writing – review & editing:** Saleh Mesbah Elkaffas.

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
