## [Decision Letter · Decision Letter 0]

6 May 2024

PONE-D-24-09209Blockchain for Video Watermarking:  An Enhanced Copyright Protection Approach for Video Forensics based on Perceptual Hash FunctionPLOS ONE

Dear Dr. Darwish,

Thank you for submitting your manuscript to PLOS ONE. After careful consideration, we feel that it has merit but does not fully meet PLOS ONE’s publication criteria as it currently stands. Therefore, we invite you to submit a revised version of the manuscript that addresses the points raised during the review process.

We look forward to receiving your revised manuscript.

Kind regards,

Brij Bhooshan Gupta

Academic Editor

PLOS ONE

A clean copy of the edited manuscript (uploaded as the new *manuscript* file).

5. We note that Figures 2, 3, 6 and 7 in your submission contain copyrighted images. All PLOS content is published under the Creative Commons Attribution License (CC BY 4.0), which means that the manuscript, images, and Supporting Information files will be freely available online, and any third party is permitted to access, download, copy, distribute, and use these materials in any way, even commercially, with proper attribution. For more information, see our copyright guidelines: http://journals.plos.org/plosone/s/licenses-and-copyright.

1. You may seek permission from the original copyright holder of Figures 2, 3, 6 and 7 to publish the content specifically under the CC BY 4.0 license.

Reviewers' comments:

Reviewer's Responses to Questions

**Comments to the Author**

1. Is the manuscript technically sound, and do the data support the conclusions?

Reviewer #1: Yes

Reviewer #2: Yes

2. Has the statistical analysis been performed appropriately and rigorously? 

Reviewer #1: Yes

Reviewer #2: No

3. Have the authors made all data underlying the findings in their manuscript fully available?

Reviewer #1: Yes

Reviewer #2: Yes

4. Is the manuscript presented in an intelligible fashion and written in standard English?

Reviewer #1: Yes

Reviewer #2: Yes

5. Review Comments to the Author

Reviewer #1: The paper "Blockchain for Video Watermarking: An Enhanced Copyright Protection Approach for Video Forensics based on Perceptual Hash Function" proposes a novel approach to video copyright protection by integrating blockchain technology with digital watermarking and a perceptual hash function. This integration aims to provide a tamper-proof, efficient, and secure method for managing video copyrights, addressing the urgent issue of copyright protection in the digital era. By storing watermark information on a blockchain structure and employing a perceptual hash function for watermark verification without the source video, the method significantly enhances the security and efficiency of copyright protection mechanisms. Experimental results demonstrate the approach's memory efficiency and robustness, making it a promising solution for copyright management in video forensics. However, while the paper provides innovative insights into using blockchain and perceptual hash functions for video watermarking, it could benefit from a more detailed exploration of the limitations and challenges associated with this integration.

• The paper mentions the use of blockchain for storing watermark information but lacks detail on the specific blockchain architecture used (e.g., public vs. private blockchain) and how blockchain's inherent properties (e.g., immutability, decentralization) are leveraged to enhance copyright protection. Could the authors elaborate on the blockchain architecture and justify their choice in the context of video watermarking?

• Section III introduces a perceptual hash function for computing hash values based on the structural information of video frames. However, the computational complexity and scalability of this hash function are not discussed. Given the potentially large size of video files and the need for real-time processing in forensic applications, how do the authors address the computational demands of the perceptual hash function?

• While the experimental section (Section IV) evaluates the approach against common video attacks, there is limited discussion on its resistance to more advanced manipulations, such as deepfake technology or sophisticated frame interpolation attacks. Can the authors provide insights or experimental evidence on the method's effectiveness against these emerging threats?

• The integration of video watermarking with blockchain raises concerns about scalability and transaction throughput, especially when dealing with high volumes of video content. How does the proposed method address the scalability challenges associated with blockchain technology, and what are the implications for the performance and responsiveness of the copyright protection system?

• Author can read the following papers to increase the technical strength of the paper:

Secure blockchain enabled Cyber–physical systems in healthcare using deep belief network with ResNet model

A Privacy-Preserving Authentic Healthcare Monitoring System Using Blockchain

Reviewer #2: 1.While the manuscript details experimental results showing memory efficiency and improved robustness due to the blockchain's random hash function, it could benefit from a more thorough comparison with existing methods. Specifically, the manuscript could include benchmark tests against traditional non-blockchain-based watermarking approaches to highlight the improvements quantitatively.

2.More detailed statistical analysis and error metrics could strengthen the validation of the experimental results.

3.The manuscript mentions the system's resilience to video frame attacks but does not deeply explore potential vulnerabilities introduced by blockchain technology itself, such as scalability issues, 51% attacks, or other common blockchain weaknesses.

4.Some assumptions are made regarding the blockchain's performance and security without detailed discussion or substantiation from real-world data or broader use cases.

5.The manuscript could include more specific details about the implementation of the perceptual hash function and the blockchain structure. For example, the type of blockchain (public, private, consortium) and the specific blockchain platform used could be specified to help replicate the study.

6.While the manuscript discusses the potential applications of the proposed system in video forensics, it could expand on the broader impact, including implications for copyright enforcement in digital media.

7.The manuscript mentions the use of performance metrics such as memory efficiency and robustness improvement, which are crucial for evaluating the effectiveness of video watermarking systems. However, it does not specify the statistical methods used to analyze these metrics, such as hypothesis testing, confidence intervals, or error analysis.

8.The manuscript could benefit from a comparative statistical analysis with existing methods. Including statistical comparisons (e.g., t-tests, ANOVA) could help highlight the improvements the proposed method offers over traditional or other blockchain-based video watermarking approaches.

6. PLOS authors have the option to publish the peer review history of their article (what does this mean?). If published, this will include your full peer review and any attached files.

Reviewer #1: No

Reviewer #2: **Yes: **Mosiur Rahaman

---

## [Author Response · Author response to Decision Letter 0]

14 Jun 2024

To The Editor-in-chief and the Associate Editor,

PLOS ONE

Manuscript ID: ONE-D-24-09209 R1

Title: Blockchain for Video Watermarking: An Enhanced Copyright Protection Approach for Video Forensics based on Perceptual Hash Function

Dear Sir,

Thanks for giving us the opportunity to revise the paper for the possible publication in your interesting journal. We also thank the reviewers for their constructive comments. We have significantly revised the manuscripts based the comments received and the details are summarized below.

Reply to the comments: 

In the revised version, what was modified and added was written in red

Reviewer 1

The paper provides innovative insights into using blockchain and perceptual hash functions for video watermarking.

Response 

Thank you for positive feedback

 It could benefit from a more detailed exploration of the limitations and challenges associated with this integration.

Response 

In the revised version, a new paragraph was added that explains in more detail the limitations and challenges associated with the integration of blockchain technology with digital watermarking and a perceptual hash function.

Integrating blockchain technology with digital watermarking and perceptual hash functions holds significant potential for enhancing content security, authenticity verification, and ownership tracking. However, this integration presents a range of limitations and challenges including: 

- Digital watermarks and perceptual hash functions generate data that needs to be stored or referenced on the blockchain. Storing large amounts of data directly on the blockchain can be expensive and impractical due to limited storage capacity and high transaction fees.

- Blockchain networks, especially those with public and permissionless configurations, often face scalability issues. The processing speed of transactions might not be fast enough to handle real-time applications of digital watermarking.

- The additional computational steps required for watermark embedding, hash generation, and blockchain transactions can introduce latency, which might not be acceptable in real-time applications.

- Ensuring compliance with data privacy regulations is crucial when storing or referencing any personal or sensitive information on the blockchain. One potential solution is to store personal data off-chain and use the blockchain to store references or hashes of the data. This way, the actual data can be deleted or modified off-chain while maintaining the integrity and immutability of the blockchain records.

Subsection V LIMITATIONS AND CHALLENGES, Pages (37&38) 

 The paper mentions the use of blockchain for storing watermark information but lacks detail on the specific blockchain architecture used (e.g., public vs. private blockchain) and how blockchain's inherent properties (e.g., immutability, decentralization) are leveraged to enhance copyright protection. Could the authors elaborate on the blockchain architecture and justify their choice in the context of video watermarking?

Response 

In the revised version, more sentences were added to specify in detail the blockchain architecture. Furthermore, the justification of our choice regarding blockchain used (public or private) and their type (immutability, or decentralization) was also mentioned. 

“Using Truffle, Solidity, and IPFS within the suggested public blockchain framework provides a robust solution for video watermarking, ensuring transparency, security, and decentralized storage. This approach leverages the strengths of public blockchains to create immutable records of video ownership and authenticity while using IPFS for efficient and decentralized storage. IPFS allows for decentralized storage of large video files, which would be impractical to store directly on the blockchain. By storing the video on IPFS and recording the hash on the blockchain, you can ensure both decentralization and efficient access”.

“Our choice of public blockchain stems from the fact that public blockchains are decentralized and maintained by multiple nodes, making it nearly impossible to alter data once it's recorded. This immutability ensures that the ownership and authenticity of a watermarked video can be reliably verified by anyone, fostering trust among content creators, distributors, and consumers. Furthermore, public blockchains are maintained by a decentralized network of nodes, reducing the risk of data being controlled or manipulated by a single entity. This decentralization can enhance security and resilience against attacks or failures; this openness increases accountability and reduces disputes over content ownership”.

Page (13)

 Section III introduces a perceptual hash function for computing hash values based on the structural information of video frames. However, the computational complexity and scalability of this hash function are not discussed. Given the potentially large size of video files and the need for real-time processing in forensic applications, how do the authors address the computational demands of the perceptual hash function?

Response 

In the revised version, new paragraphs were added to discuss the scalability issue of utilizing the perceptual hash function for computing hash values based on the structural information of video frames. Furthermore, the computational complexity of building this type of hash was also highlighted. 

“Using perceptual hash functions for video watermarking combined with blockchain technology can significantly enhance content verification and integrity. Addressing scalability challenges involves efficient hashing algorithms, segment-based hashing, and off-chain storage solutions like IPFS. In our suggested model, we utilize a simplified version of pHash that can reduce computational overhead to generate compact hash values quickly. Furthermore, instead of generating perceptual hashes for every frame, divide the video into segments (e.g., every few seconds or key frames) and generate a hash for each segment. This reduces the number of hashes while still providing adequate verification. Finally, using IPFS to store the perceptual hashes and video files off-chain using decentralized storage solutions like IPFS. Only store the IPFS hash (a unique identifier) on the blockchain, which points to the actual data stored off-chain. By implementing these strategies, it is possible to achieve a scalable and robust system for managing and verifying video content in a decentralized manner”.

“The computational complexity of building perceptual hash functions for video watermarking is an important factor to consider, especially for scalability and real-time applications. This process generally depends on the specific algorithm used and the size of the video content. Its complexity is O(n log⁡n) for the for the DCT computation, where n is the number of pixels in the resized image/frame. So the total complexity is O(F×H), where F is the number of frames and H is the complexity of the perceptual hash function”.

Pages (17& 18)

 While the experimental section (Section IV) evaluates the approach against common video attacks, there is limited discussion on its resistance to more advanced manipulations, such as deep fake technology or sophisticated frame interpolation attacks. Can the authors provide insights or experimental evidence on the method's effectiveness against these emerging threats?

Response 

In the revised version, new paragraphs were added to highlight the suggested model’s resistance to more advanced manipulations, such as deepfake technology or sophisticated frame interpolation attack.

“Regarding the suggested model’s resistance to more advanced manipulations, such as deepfake technology or sophisticated frame interpolation attack. Deepfake algorithms primarily focus on the luminance channel to ensure high visual fidelity and realism in terms of shapes and edges [56]. In our suggested model, embedding watermarks in the chrominance channel can therefore survive these manipulations better than those in the luminance channel. Deepfake algorithms might not accurately replicate the subtleties of the chrominance channels, watermarks embedded in these channels can help in detecting inconsistencies and manipulations. By comparing the extracted watermark from a potentially deepfaked video with the original, it is possible to identify tampering. Also, as our model embeds the watermark in the frequency domain; it can make the watermark more resilient to deepfake manipulations [57].”

“Frame interpolation attacks involve generating intermediate frames between existing frames in a video to create smooth transitions, often using techniques like deep learning-based algorithms. This type of manipulation can distort or disrupt embedded watermarks. Our strategies for enhancing watermark robustness against frame interpolation rely on distributing the watermark information across multiple frames instead of embedding it in single frames. This way, even if some frames are interpolated or modified, the watermark can still be reconstructed from the remaining frames. Furthermore, adjust the watermark embedding process based on the content of each frame. This approach can use characteristics like motion vectors or scene changes to decide where and how to embed the watermark, making it more resilient to interpolation artifacts [58].”

Pages (33&34&35)

Furthermore, new sentences were added in the future work section for potential directions for further research and development of our work regarding its resilience to deepfake attacks. 

“Currently, we haven't conducted a dedicated evaluation of the model's resistance against deepfakes. However, based on the watermark placement strategy, we believe the impact would be limited in most cases. Further investigation using deepfake datasets specifically designed to target watermarked content is necessary to fully assess the model's robustness in this scenario.”

Page (39)

 The integration of video watermarking with blockchain raises concerns about scalability and transaction throughput, especially when dealing with high volumes of video content. How does the proposed method address the scalability challenges associated with blockchain technology, and what are the implications for the performance and responsiveness of the copyright protection system?

Response 

In the revised version, new paragraphs were added that highlight how the suggested model addresses the scalability challenges associated with blockchain technology. Also, the implications for the performance and responsiveness of the copyright protection system were discussed in detail.

“The integration of video watermarking with blockchain offers a secure and tamper-proof solution for copyright protection. However, a major concern arises with scalability, particularly when dealing with large volumes of video content. The proposed method effectively addresses these challenges through IPFS storage and Merkle DAGs. Instead of storing the entire watermarked video directly on the blockchain, the system utilizes IPFS for efficient storage and retrieval of video data. IPFS's distributed storage architecture and content-based addressing mechanisms provide a scalable and cost-effective solution for handling large video files (off-chain watermark storage)”.

“Furthermore, to maintain data integrity while reducing blockchain storage requirements, the proposed method employs Merkle DAGs (directed acyclic graphs) to store hashes of the watermarked video content on the blockchain. Merkle DAGs allow for more efficient updates and verification of data compared to traditional Merkle trees. This is particularly beneficial for video watermarking, where new watermarks may be added or existing ones updated frequently. Additionally, Merkle DAGs are immutable, meaning once a hash is added, it cannot be altered. This ensures the integrity of the watermark data and prevents unauthorized modifications. Finally, Merkle DAGs provide greater flexibility in organizing and structuring data, making them well-suited for handling complex data relationships like those encountered in video watermarking”.

Page (24)

Yet, the use of blockchain for video watermarking in a copyright protection system can significantly influence both performance and responsiveness. Here are the key implications: (1) as more videos are watermarked and registered on the blockchain, the size of the blockchain grows. This can affect the speed of transaction validation and block creation. (2) Blockchain transactions, especially in public blockchains, can have a high latency due to the time required for block confirmation. This can delay the process of watermark registration and verification. (3) Blockchain's decentralized nature means that every transaction needs to be confirmed by the network, which can take time. This can affect the ability to perform real-time watermark verification. (4) The responsiveness of the system can vary based on the number and distribution of nodes. More nodes can mean more security but also potentially more latency.

By effectively utilizing IPFS and Merkle DAGs, the proposed method demonstrates a well-considered approach to addressing scalability challenges in blockchain-based video watermarking. The strategies employed were to minimize storage overhead, improve transaction throughput, and optimize performance for handling large volumes of video content. The use of Merkle DAGs in particular provides additional advantages in terms of immutability, efficiency, and flexibility, making it a suitable choice for securing and managing video watermark data in a scalable and secure manner.

Pages(27&28) 

 Author can read the following papers to increase the technical strength of the paper: 

Secure blockchain enabled Cyber–physical systems in healthcare using deep belief network with ResNet model 

A Privacy-Preserving Authentic Healthcare Monitoring System Using Blockchain

Response

Thank you very much for helping, the two paper were cited in the manuscript. Refs. [59]&[60]

Reviewer 2

 While the manuscript details experimental results showing memory efficiency and improved robustness due to the blockchain's random hash function, it could benefit from a more thorough comparison with existing methods. Specifically, the manuscript could include benchmark tests against traditional non-blockchain-based watermarking approaches to highlight the improvements quantitatively.

Response

In the revised version, a new set of experiments was conducted to compare the suggested blockchain-based video watermarking technique with traditional non-blockchain-based watermarking approaches to highlight the improvements quantitatively. 

The proposed blockchain–based technique has been compared with traditional two recent non-blockchain video-watermarking techniques presented in Refs. [61][62]. In Ref.[61], a lossless video watermarking system is suggested that operates by selecting keyframes optimally via the use of a linear wavelet transform and an intelligent gravitational search algorithm. Using the histogram difference approach, this methodology extracts color motion as well as static frames from the cover video. The chrominance channel of the motion frames undergoes a one-level linear wavelet transform, and a low-frequency sub-band LL is chosen for the watermark embedding. In Ref. [62], a secure video watermarking method is provided employing a novel two-dimensional complex map. In order to demonstrate the chaotic nature of the suggested map, standard analyzes have been done on a dynamical system. There is also an efficient approach for embedding and extraction watermark that uses single value decomposition in conjunction with the IWT, DWT, and CT transforms. 

This set of experiments measures the imperceptibility performance of the proposed technique against existing techniques in terms of mean peak signal to-noise ratio (MPSNR) and Mean Structural Similarity Index Measure (MSSIM) under different video attacks. See [61] for more details regarding these metrics. Table 5 highlights the MPSNR and MSSIM values corresponding to each video against compared schemes. It can be noticed from the table that the proposed technique attains an av

---

## [Decision Letter · Decision Letter 1]

24 Jul 2024

Blockchain for Video Watermarking:  An Enhanced Copyright Protection Approach for Video Forensics based on Perceptual Hash Function

PONE-D-24-09209R1

Dear Dr. Darwish,

We’re pleased to inform you that your manuscript has been judged scientifically suitable for publication and will be formally accepted for publication once it meets all outstanding technical requirements.

Kind regards,

Brij Bhooshan Gupta

Academic Editor

PLOS ONE

Additional Editor Comments (optional):

Reviewers' comments:

Reviewer's Responses to Questions

**Comments to the Author**

1. If the authors have adequately addressed your comments raised in a previous round of review and you feel that this manuscript is now acceptable for publication, you may indicate that here to bypass the “Comments to the Author” section, enter your conflict of interest statement in the “Confidential to Editor” section, and submit your "Accept" recommendation.

Reviewer #1: All comments have been addressed

Reviewer #2: All comments have been addressed

2. Is the manuscript technically sound, and do the data support the conclusions?

Reviewer #1: Yes

Reviewer #2: Yes

3. Has the statistical analysis been performed appropriately and rigorously? 

Reviewer #1: (No Response)

Reviewer #2: Yes

4. Have the authors made all data underlying the findings in their manuscript fully available?

Reviewer #1: (No Response)

Reviewer #2: Yes

5. Is the manuscript presented in an intelligible fashion and written in standard English?

Reviewer #1: (No Response)

Reviewer #2: Yes

6. Review Comments to the Author

Reviewer #1: The authors have made significant changes in the manuscript. So, it should be accepted.

The authors have made significant changes in the manuscript. So, it should be accepted.

The authors have made significant changes in the manuscript. So, it should be accepted.

Reviewer #2: Authors have appropriately addressed every comment made in a past review, hence I believe this work to be now suitable for publication. From my perspective, no more review.

7. PLOS authors have the option to publish the peer review history of their article (what does this mean?). If published, this will include your full peer review and any attached files.

Reviewer #1: No

Reviewer #2: **Yes: **Mosiur Rahaman

---

## [Editor Report · Acceptance letter]

22 Aug 2024

PONE-D-24-09209R1 

PLOS ONE

Dear Dr. Darwish, 

I'm pleased to inform you that your manuscript has been deemed suitable for publication in PLOS ONE. Congratulations! Your manuscript is now being handed over to our production team.

Kind regards, 

on behalf of

Dr. Brij Bhooshan Gupta 

Academic Editor

PLOS ONE